# LLMLingua: Compressing Prompts for Accelerated Inference of Large Language Models

**Huiqiang Jiang, Qianhui Wu, Chin-Yew Lin, Yuqing Yang, Lili Qiu**
Microsoft Corporation
{hjiang, qianhuiwu, cyl, yuqing.yang, liliqiu}@microsoft.com

## Abstract

Large language models (LLMs) have been applied in various applications due to their astonishing capabilities. With advancements in technologies such as chain-of-thought (CoT) prompting and in-context learning (ICL), the prompts fed to LLMs are becoming increasingly lengthy, even exceeding tens of thousands of tokens. To accelerate model inference and reduce cost, this paper presents *LLMLingua*, a coarse-to-fine prompt compression method that involves a budget controller to maintain semantic integrity under high compression ratios, a token-level iterative compression algorithm to better model the interdependence between compressed contents, and an instruction tuning based method for distribution alignment between language models. We conduct experiments and analysis over four datasets from different scenarios, *i.e.*, GSM8K, BBH, ShareGPT, and Arxiv-March23; showing that the proposed approach yields state-of-the-art performance and allows for up to 20x compression with little performance loss.[1]

## 1 Introduction

The widespread adoption of ChatGPT has transformed numerous scenarios by harnessing the powerful generalization and reasoning capabilities of large language models (LLMs). In practical applications, crafting suitable prompts is crucial and usually involves techniques such as chain-of-thought, in-context learning, and retrieving related documents or historical conversations (Wei et al., 2022; Chase, 2022). While these methods can elicit highly effective generations by activating LLMs' domain-specific knowledge, they often require longer prompts. Therefore, striking a balance between the massive computational demands of LLMs and the need for longer prompts has become an urgent issue. Some studies attempt to accelerate model inference by modifying the parameters of LLMs through quantization (Dettmers et al., 2022; Xiao et al., 2023), compression (Frantar and Alistarh, 2023), etc. However, these approaches may be not suitable when the LLMs can be accessed via APIs only.

Approaches that attempt to reduce the length of original prompts while preserving essential information have emerged lately. These approaches are grounded in the concept that natural language is inherently redundant (Shannon, 1951) and thus can be compressed. Gilbert et al. (2023) also indicate that LLMs can effectively reconstruct source code from compressed text descriptions while maintaining a high level of functional accuracy. Therefore, we follow this line of studies to compress a long prompt into a shorter one without any gradient flow through the LLMs to support applications based on a larger range of LLMs.

In terms of information entropy, tokens with lower perplexity (PPL) contribute less to the overall entropy gains of the language model. In other words, removing tokens with lower perplexity has a relatively minor impact on the LLM's comprehension of the context. Motivated by this, Li (2023) propose Selective-Context, which first employs a small language model to compute the self-information of each lexical unit (such as sentences, phrases, or tokens) in original prompts, and then drops the less informative content for prompt compression. However, this method not only ignores the interdependence between the compressed contents but also neglects the correspondence between the LLM being targeted and the small language model used for prompt compression.

This paper proposes *LLMLingua*, a coarse-to-fine prompt compression method, to address the aforementioned issues. Specifically, we first present a budget controller to dynamically allocate different compression ratios to various components in original prompts such as the instruction, demonstrations, and the question, and meanwhile,

---

[1]Our code is available at https://aka.ms/LLMLingua.

perform coarse-grained, demonstration-level compression to maintain semantic integrity under high compression ratios. We further introduce a token-level iterative algorithm for fine-grained prompt compression. Compared with *Selective Context*, it can better preserve the key information within the prompt by taking into account the conditional dependencies between tokens. Additionally, we pose the challenge of distribution discrepancy between the target LLM and the small language model used for prompt compression, and further propose an instruction tuning based method to align the distribution of both language models.

We validate the effectiveness of our approach on four datasets from different domains, *i.e.*, GSM8K and BBH for reasoning and ICL, ShareGPT for conversation, and Arxiv-March23 for summarization. The results show that our method yields state-of-the-art performance across the board. Furthermore, we conduct extensive experiments and discussions to analyze why our approach attains superior performance. To our best knowledge, we are the first to evaluate reasoning and ICL capabilities in the domain of efficient LLMs.

## 2 Related Work

### 2.1 Efficient LLMs

Efficient large language models have gained significant attention in recent research community, especially with the growing prominence of Chat-GPT. Most of these methods aim to reduce the costs of inference and fine-tuning by modifying the model parameters through quantization (Dettmers et al., 2022; Frantar et al., 2023; Xiao et al., 2023), compression (Frantar and Alistarh, 2023), instruct tuning (Taori et al., 2023; Chiang et al., 2023; Xu et al., 2023), or delta tuning (Hu et al., 2022).

A line of studies attempt to optimize inference costs from the perspective of the input prompts. Motivated by the observation of the abundance of identical text spans between the input and the generated result, Yang et al. (2023) directly copy tokens from prompts for decoding to accelerate the inference process of LLMs. Some approaches focus on compressing prompts, specifically, learning special tokens via prompt tuning of LLMs to reduce the number of tokens to be processed during inference (Mu et al., 2023; Ge et al., 2022; Wingate et al., 2022; Chevalier et al., 2023; Ge et al., 2023). Unfortunately, these methods are usually tailored to particular tasks and some of them (Mu et al., 2023;

Chevalier et al., 2023) even require to fine-tune the whole language model, which severely limits their application scenarios. Furthermore, there are some studies (Chase, 2022; Zhang et al., 2023) that attempt to utilize LLMs to summarize dialog or data, thereby forming memory and knowledge. However, these approaches require multiple invocations of LLMs, which are quite costly.

Some methods reduce the prompt length by selecting a subset of demonstrations. For example, Zhou et al. (2023) introduces a reinforcement learning based algorithm to allocate a specific number of demonstrations for each question. Some other methods focus on token pruning (Goyal et al., 2020; Kim and Cho, 2021; Kim et al., 2022; Rao et al., 2021; Modarressi et al., 2022) and token merging (Bolya et al., 2023). However, these approaches are proposed for smaller models such as BERT, ViT. Moreover, they depend on fine-tuning the models or obtaining intermediate results during inference.

The most similar work to this paper is Selective-Context (Li, 2023), which evaluates the informativeness of lexical units by computing self-information with a small language model, and drops the less informative content for prompt compression. This paper is inspired by Selective-Context and further proposes a coarse-to-fine framework to address its limitations.

### 2.2 Out-of-Distribution (OoD) Detection

Recently, a series of studies have been proposed for unsupervised OoD detection. With only in-distribution texts available for learning, these methods either fine-tune a pre-trained language model (Arora et al., 2021) or train a language model from scratch (Mai et al., 2022). Wu et al. (2023) analyze the characteristics of these methods and leverage multi-level knowledge distillation to integrate their strengths while mitigating their limitations. Finally, perplexity output by the resulting language model is used as the indication of an example being OoD.

This paper also regards perplexity as a measurement of how well a language model predicts a sample. In contrast to out-of-distribution detection, which identifies examples with high perplexities as indicative of unreliable predictions, we consider tokens with higher perplexity to be more influential during the inference process of language models.

### 2.3 LLMs as a Compressor

Recently, many perspectives have interpreted large language models and unsupervised learn-

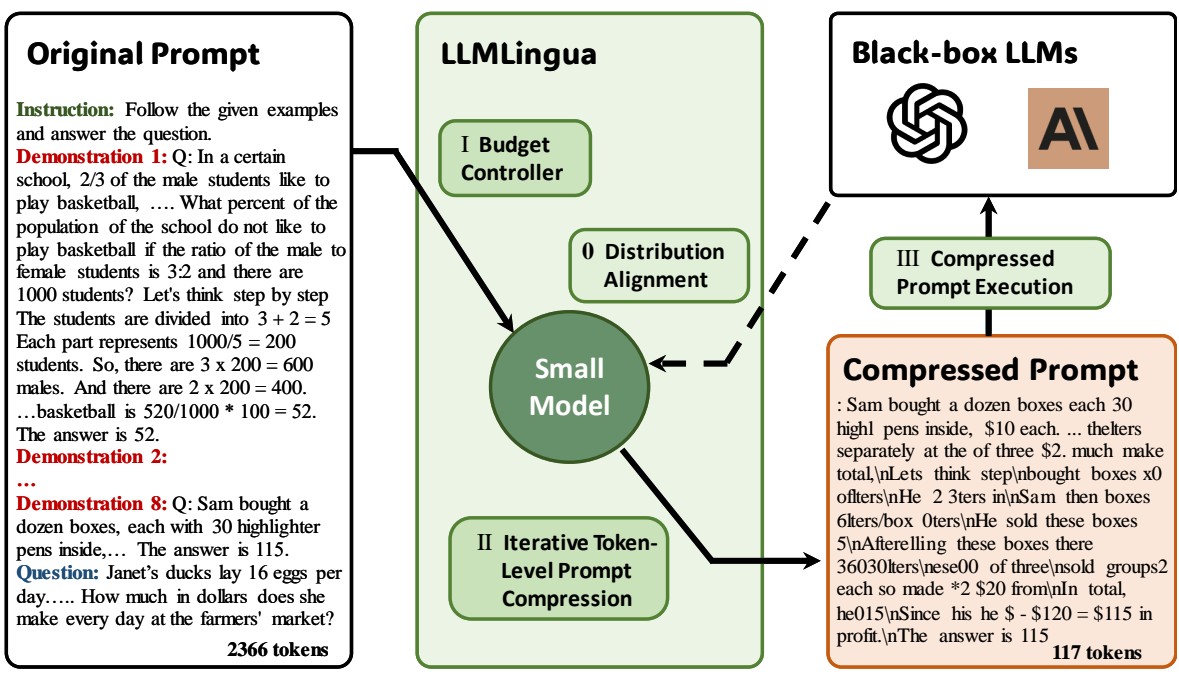

**Original Prompt**

**Instruction:** Follow the given examples and answer the question.
**Demonstration 1:** Q: In a certain school, 2/3 of the male students like to play basketball, …. What percent of the population of the school do not like to play basketball if the ratio of the male to female students is 3:2 and there are 1000 students? Let's think step by step The students are divided into 3 + 2 = 5 Each part represents 1000/5 = 200 students. So, there are 3 x 200 = 600 males. And there are 2 x 200 = 400. …basketball is 520/1000 * 100 = 52. The answer is 52.
**Demonstration 2:**
…
**Demonstration 8:** Q: Sam bought a dozen boxes, each with 30 highlighter pens inside,… The answer is 115.
**Question:** Janet's ducks lay 16 eggs per day….. How much in dollars does she make every day at the farmers' market?
**2366 tokens**

**LLMLingua**

I **Budget Controller**

0 **Distribution Alignment**

**Small Model**

II **Iterative Token-Level Prompt Compression**

**Black-box LLMs**

III **Compressed Prompt Execution**

**Compressed Prompt**

: Sam bought a dozen boxes each 30 highl pens inside, $10 each. ... thelters separately at the of three $2. much make total,\nLets think step\nbought boxes x0 oflters\nHe 2 3ters in\nSam then boxes 6lters/box 0ters\nHe sold these boxes 5\nAfterelling these boxes there 36030lters\nese00 of three\nsold groups2 each so made *2 $20 from\nIn total, he015\nSince his he $ - $120 = $115 in profit.\nThe answer is 115
**117 tokens**

Figure 1: Framework of the proposed approach *LLMLingua*.

ing as a kind of compressor for world knowledge (Sutskever, 2023; Delétang et al., 2023), by using arithmetic coding (Rissanen, 1976; Pasco, 1976). Our research can be viewed as an endeavor to further compress information within prompts by capitalizing on the compression-like characteristics of large language models.

## 3 Problem Formulation

A prompt compression system is designed to generate a compressed prompt $\widetilde{x} = \{\widetilde{x}_i\}_{i=1}^{\widetilde{L}}$ from a given original prompt $x = (x^{\text{ins}}, x^{\text{dems}}, x^{\text{que}})$, where $x^{\text{ins}} = \{x_i^{\text{ins}}\}_{i=1}^{L^{\text{ins}}}$, $x^{\text{dems}} = \{x_i^{\text{dems}}\}_{i=1}^{L^{\text{dems}}}$, and $x^{\text{que}} = \{x_i^{\text{que}}\}_{i=1}^{L^{\text{que}}}$ denote the instruction, demonstrations, and the question in the original prompt $x$. $\widetilde{L}$, $L_{\text{ins}}$, $L_{\text{dems}}$, and $L_{\text{que}}$ represent the numbers of tokens in $\widetilde{x}$, $x^{\text{ins}}$, $x^{\text{dems}}$, and $x^{\text{que}}$, respectively. Let $L = L_{\text{ins}} + L_{\text{dems}} + L_{\text{que}}$ denote the total sequence length of $x$, the compression rate is defined as $\tau = \widetilde{L}/L$, $\tau \in [0, 1]$, and the compression ratio is $1/\tau$. A smaller value of $\tau$ implies a lower inference cost, which is preferable. Let $\widetilde{x}_G$ represent the LLM-generated results derived by $\widetilde{x}$ and $x_G$ denotes the tokens derived by $x$, the distribution of $\widetilde{x}_G$ is expected to be as similar to $x_G$ as possible. This can be formulated as:

$$\min_{\widetilde{x}, \tau} \text{KL}(P(\widetilde{x}_G|\widetilde{x}), P(x_G|x)), \quad (1)$$

## 4 Methodology

In this section, we elaborate on the proposed coarse-to-fine prompt compression approach, *LLMLingua*. First, we introduce a budget controller to dynamically allocate different compression ratios to various components in prompts and meanwhile, perform coarse-grained, demonstration-level compression to maintain semantic integrity under high compression ratios. Next, we describe the proposed iterative prompt algorithm designed to retain knowledge from the prompt while compressing. Finally, we introduce alignment to address the distribution gap between the small model and black-box large models. Figure 1 show the framework.

### 4.1 Budget Controller

The budget controller here is designed to allocate different budgets, *i.e.*, compression ratio, to different components in a prompt such as instructions, demonstrations, and questions, at the sentence or demonstration level. There are two considerations:

(i) In general, the instruction and the question in a prompt have a direct influence on the generated results, as they should contain all the necessary knowledge to generate the following answer. On the contrary, if there are multiple demonstrations in the original prompt, the conveyed information may be redundant. Therefore, a tailored budget controller is required to allocate more budget (*i.e.*,

**Algorithm 1** Pseudo code of Budget Controller.

**Input**: A small language model $\mathcal{M}_s$; the original prompt $\boldsymbol{x} = (\boldsymbol{x}^{\text{ins}}, \boldsymbol{x}^{\text{dems}}, \boldsymbol{x}^{\text{que}})$.
1: Set the selected demonstration set $\mathcal{D} = \phi$.
2: Get demonstration compression rate $\tau_{\text{dem}}$ by Eq.(2).
3: Calculate the perplexity of each demonstration via $\mathcal{M}_s$.
4: Rank all demonstrations in descending order of their perplexity as a list $(\boldsymbol{x}^{\text{dem}}_{(1)}, ..., \boldsymbol{x}^{\text{dem}}_{(N)})$, where $N$ is the number of demonstrations, $\boldsymbol{x}^{\text{dem}}_{(i)}$ is the $i$-th demonstration.
5: **for** $i = 1$ **do**
6:     **if** $\widetilde{L}_{\mathcal{D}} > k \cdot \tau_{\text{dems}} L_{\text{dems}}$ **then**
7:         Break.
8:     **end if**
9:     Append $\boldsymbol{x}^{\text{dem}}_{(i)}$ to $\mathcal{D}$.
10:     $i = i + 1$
11: **end for**
12: Allocate remaining budget to $\boldsymbol{x}^{\text{ins}}$ and $\boldsymbol{x}^{\text{que}}$ via Eq. (3).

**Output**: The subset of demonstrations $\mathcal{D}$ obtained from coarse-grained compression; Additional budget $\Delta\tau_{\text{ins,que}}$ for the instruction and the question.

---

**Algorithm 2** Pseudo code of Iterative Token-level Prompt Compression (ITPC).

**Input**: A small language model $\mathcal{M}_s$; the prompt from budget controller $\boldsymbol{x}' = (\boldsymbol{x}^{\text{ins}}, \boldsymbol{x}^{\mathcal{D}}, \boldsymbol{x}^{\text{que}})$; target compression rate $\tau$, adjusted compression rate $\triangle\tau_{\text{ins,que}}$.
1: Set the selected token set $\mathcal{T} = \phi$
2: Get segment set $\mathcal{S}$.
3: **for** $i = 1, 2, \ldots, m$ **do**
4:     Get the conditional probabilities $p(\boldsymbol{s}_i)$ via Eq.(5)
5:     Get the compression threshold $\gamma_i$ with Eq. (6).
6:     Append the compressed token to $\mathcal{T}$ via Eq.(7).
7: **end for**
8: Concatenate all tokens in $\mathcal{T}$ as $\widetilde{\boldsymbol{x}}$.

**Output**: The compressed prompt $\widetilde{\boldsymbol{x}}$.

---

smaller compression ratios) for instructions and questions, and less budget for demonstrations.

(ii) When a high compression ratio is required, token-level dropout as in Li (2023) might make the compressed prompts too trivial and thus lose vital information from the original prompt. Consequently, sentence-level dropout should be employed instead to preserve a certain degree of linguistic integrity. Especially in the case of multiple redundant demonstrations, we can even perform demonstration-level control to meet the compression requirement.

Algorithm 1 illustrates the overall procedure of the budget controller.

**Derive compression ratio for demonstrations.** We first compute the compression rate for demonstrations $\tau_{\text{dems}}$ according to the target overall compression rate $\tau$ and the pre-defined compression rate for instructions and questions, *i.e.*, $\tau_{\text{ins}}$ and $\tau_{\text{que}}$, respectively.

$$\tau_{\text{dems}} = \frac{\tau L - (\tau_{\text{ins}} L_{\text{ins}} + \tau_{\text{que}} L_{\text{que}})}{L_{\text{dems}}}. \quad (2)$$

**Demonstration-level prompt compression.** With the derived $\tau_{\text{dems}}$ for demonstrations, we then perform a coarse-grained demonstration-level prompt compression: we construct $\mathcal{D}$, a subset of demonstrations from $\boldsymbol{x}^{\text{dems}}$.

Specifically, we first employ a small language model $\mathcal{M}_s$, such as GPT-2 or LLaMA, to compute the perplexity of each demonstration in $\boldsymbol{x}^{\text{dems}}$. Then, we select demonstrations in descending order of their perplexity values, until adding one more

demonstration to $\mathcal{D}$ will make the total number of tokens in $\mathcal{D}$ exceed maximum tokens $k \cdot \tau_{\text{dems}} L_{\text{dems}}$, where $k$ is the granular control coefficient.

**Adjust compression ratios for instruction and question.** After obtaining the coarse-grained compression result $\mathcal{D} = \{x_i\}_{i=1}^{\widetilde{L}_{\mathcal{D}}}$, we allocate the remaining budget to the instruction and the question:

$$\Delta\tau = \frac{k \cdot \tau_{\text{dems}} L_{\text{dems}} - \widetilde{L}_{\mathcal{D}}}{L_{\text{ins}} + L_{\text{que}}}, \quad (3)$$

where $\widetilde{L}_{\mathcal{D}}$ denote the total number of tokens in $\mathcal{D}$.

### 4.2 Iterative Token-level Prompt Compression

Utilizing perplexity for prompt compression encounters the intrinsic limitation, *i.e.*, the independence assumption, similar to the shortcomings of the Mask Language Model (Yang et al., 2019) as:

$$
\begin{aligned}
p(\widetilde{\boldsymbol{x}}) &= \prod_{i=1}^{\widetilde{L}} p(\widetilde{x}_i | \widetilde{x}_{<i}) \\
&\approx p(\boldsymbol{x}') = \prod_{i=1}^{L'} p(x_i | \widetilde{x}_{<i}, \overline{x}_{<i}),
\end{aligned} \quad (4)
$$

where $\boldsymbol{x}' = (\boldsymbol{x}^{\text{ins}}, \boldsymbol{x}^{\mathcal{D}}, \boldsymbol{x}^{\text{que}})$ is the original prompt after demonstration-level compression; $\boldsymbol{x}^{\mathcal{D}}$ is the concatenation of all demonstrations in $\mathcal{D}$; $\widetilde{x}$ is the final compressed prompt; $\widetilde{x}_{<i}$ and $\overline{x}_{<i}$ denote the preserved and compressed tokens before the $i$-th token $x_i$; $L'$ and $\widetilde{L}$ denote the numbers of all tokens in $\boldsymbol{x}'$ and $\widetilde{\boldsymbol{x}}$, respectively.

Here we propose an iterative token-level prompt compression (ITPC) algorithm to mitigate the inaccuracy introduced by the conditional independence assumption. Algorithm 2 shows the pseudo codes.

Specifically, we first divide the target prompt $\boldsymbol{x}'$ into several segments $\mathcal{S} = \{\boldsymbol{s}_1, \boldsymbol{s}_2, ..., \boldsymbol{s}_m\}$. And

then, we use the smaller model $\mathcal{M}_s$ to obtain the perplexity distribution of all segments. The compressed prompt obtained from each segment is concatenated to the subsequent segment, enabling more accurate estimation of the conditional probability. The corresponding probability estimation function can be formulated as:

$$
\begin{aligned}
p(\widetilde{\boldsymbol{s}}_j) &= \prod_{i=1}^{\sum_k^j \widetilde{L}_{s,k}} p(\widetilde{s}_{j,i}|\widetilde{s}_{j,<i}, \widetilde{\boldsymbol{s}}_{<j}) \\
&\approx \prod_{i=1}^{L_{s,j}+\sum_k^{j-1} \widetilde{L}_{s,k}} p(s_{j,i}|s_{j,<i}, \widetilde{\boldsymbol{s}}_{<j}),
\end{aligned}
\tag{5}
$$

where $s_{j,i}$ denotes the $i$-th token in the $j$-th segment, $L_{s,j}$ and $\widetilde{L}_{s,j}$ represent the token length of $j$-th original and compressed segment, respectively.

When the conditional probabilities for each segment $p(\boldsymbol{s}_j)$ are obtained, the compression ratio threshold $\gamma_j$ w.r.t. $\boldsymbol{s}_j$ are dynamically calculated based on the PPL distribution and the corresponding compression ratio $\tau_{\boldsymbol{s}_j}$, where

$$
\tau_{\boldsymbol{s}_j} = \begin{cases} \tau_{\text{ins}} + \Delta\tau, & \text{if } \boldsymbol{s}_j \text{ from } \boldsymbol{x}^{\text{ins}}, \\ \tau_{\text{dems}}, & \text{if } \boldsymbol{s}_j \text{ from } \boldsymbol{x}^{\mathcal{D}}, \\ \tau_{\text{que}} + \Delta\tau, & \text{if } \boldsymbol{s}_j \text{ from } \boldsymbol{x}^{\text{que}}. \end{cases}
\tag{6}
$$

Finally, tokens in each $\boldsymbol{s}_j$ with the PPL greater than $\gamma_j$ are retained in the compressed prompt.

$$
\widetilde{\boldsymbol{s}}_j = \{s_{j,i}|p(s_{j,i}) > \gamma_j\}
\tag{7}
$$

### 4.3 Distribution Alignment

To narrow the gap between the distribution of the LLM and that of the small language model used for prompt compression, here we align the two distributions via instruction tuning.

Specifically, we start from a pre-trained small language model $\mathcal{M}_s$ and use the data generated by the LLM to perform instruction tuning on $\mathcal{M}_s$. The optimization of $\mathcal{M}_s$ can be formulated as:

$$
\min_{\boldsymbol{\theta}_s} \mathbb{E}\left[\frac{1}{N}\sum_{i=1}^{N} \mathcal{L}\left(\mathbf{x}_i, \mathbf{y}_{i,\text{LLM}}; \boldsymbol{\theta}_{\mathcal{M}_s}\right)\right],
\tag{8}
$$

where $\theta_{\mathcal{M}_s}$ denotes the parameters of $\mathcal{M}_s$, $(\boldsymbol{x}_i, \boldsymbol{y}_i^{\text{LLM}})$ denotes the pair of instruction $\boldsymbol{x}_i$ and the LLM generated texts $\boldsymbol{y}_i^{\text{LLM}}$, $N$ is the number of all examples used for instruction tuning.

## 5 Experiments

### 5.1 Settings

**Datasets** To comprehensively assess the effectiveness of compressed prompts in retaining LLM abilities, we evaluated their performance across four datasets. For reasoning and in-context learning (ICL), we use **GSM8K** (Cobbe et al., 2021) and **BBH** (Suzgun et al., 2022). As for contextual understanding, we use **ShareGPT** (sha, 2023) for conversation and **Arxiv-March23** (Li, 2023) for summarization. It's worth noting that neither the small LM nor the target LLMs used in this paper have seen any of the evaluation datasets, especially the last two which were newly collected this year. We followed the experimental setup of previous work (Fu et al., 2023a; Li, 2023) for the usage of these datasets. Please refer to Appendix A.1 for detailed information.

**Evaluation** Following Cobbe et al. (2021), Fu et al. (2023a), and Li (2023), we utilize the Exact Match as the evaluation metric for GSM8K and BBH. We use BLEU (Papineni et al., 2002), ROUGE (Lin, 2004), and BERTScore (Zhang et al., 2020) as the evaluation metrics for ShareGPT and Arxiv-March23.

**Implementation Details** In this paper, we employ the GPT-3.5-Turbo-0301 and the Claude-v1.3 as the target LLMs, which can be accessed via OpenAI[2] and Claude API[3]. To improve the stability of outputs produced by LLMs we apply greedy decoding with a temperature of 0 across all experiments. The Alpaca dataset (Taori et al., 2023) is exclusively employed for aligning small language models with black-box LLMs, and is not utilized in the evaluation process. In our experiments, we utilize either Alpaca-7B[4] or GPT2-Alpaca as the small pre-trained language model $\mathcal{M}_s$ for compression. We implement our approach based on PyTorch 1.12.0[5] and Huggingface's Transformers[6]. We set the granular control coefficient $k$ to 2. We use the pre-defined compression rates $\tau_{\text{ins}} = 0.85$ and $\tau_{\text{que}} = 0.9$ for instructions and questions. The segment size used in the iterative token-level compression is set to 100.

---

[2]https://platform.openai.com/
[3]https://anthropic.com/
[4]https://github.com/tatsu-lab/stanford_alpaca
[5]https://pytorch.org/
[6]https://github.com/huggingface/transformers

| Methods | ShareGPT | | | | | | | Arxiv-March23 | | | | | | |
|---|---|---|---|---|---|---|---|---|---|---|---|---|---|---|
| | BLEU | Rouge1 | Rouge2 | RougeL | BS F1 | Tokens | $1/\tau$ | BLEU | Rouge1 | Rouge2 | RougeL | BS F1 | Tokens | $1/\tau$ |
| *Constraint I* | *2x constraint* | | | | | | | *350 tokens constraint* | | | | | | |
| Sentence Selection | **28.59** | 46.11 | **31.07** | 37.94 | 88.64 | 388 | 1.5x | 22.77 | 50.1 | 25.93 | 33.63 | 88.21 | 379 | 4x |
| Selective-Context | 25.42 | 46.47 | 29.09 | 36.99 | 88.92 | 307 | 1.9x | 21.41 | 51.3 | 27.94 | 36.73 | 89.60 | 356 | 4x |
| **Ours** | 27.36 | **48.87** | 30.32 | **38.55** | **89.52** | 304 | 1.9x | **23.15** | **54.21** | **32.66** | **42.74** | **90.33** | 345 | 4x |
| *Constraint II* | *3x constraint* | | | | | | | *175 tokens constraint* | | | | | | |
| Sentence Selection | 18.94 | 35.17 | 18.96 | 26.75 | 85.63 | 255 | 2.3x | 12.41 | 38.91 | 14.25 | 26.72 | 87.09 | 229 | 7x |
| Selective-Context | 15.79 | 38.42 | 20.55 | 28.89 | 87.12 | 180 | 3.3x | 12.23 | 42.47 | 19.48 | 29.47 | 88.16 | 185 | 8x |
| **Ours** | **19.55** | **40.81** | **22.68** | **30.98** | **87.70** | 177 | 3.3x | **13.45** | **44.36** | **24.86** | **34.94** | **89.03** | 176 | 9x |

Table 1: Performance of different methods under different target compression ratios on the conversation (ShareGPT) and summarization (Arxiv-March23) task.

| Methods | GSM8K | | | BBH | | |
|---|---|---|---|---|---|---|
| | EM | Tokens | $1/\tau$ | EM | Tokens | $1/\tau$ |
| Full-shot | 78.85 | 2,366 | - | 70.07 | 774 | - |
| *1-shot constraint* | | | | | | |
| 1-shot | 77.10 | 422 | 6x | 69.60 | 284 | 3x |
| Selective-Context | 53.98 | 452 | 5x | 54.27 | 276 | 3x |
| GPT4 Generation | 71.87 | 496 | 5x | 27.13 | 260 | 3x |
| **Ours** | **79.08** | 446 | 5x | **70.11** | 288 | 3x |
| *half-shot constraint* | | | | | | |
| Sentence Selection | 72.33 | 230 | 10x | 39.56 | 175 | 4x |
| Selective-Context | 52.99 | 218 | 11x | 54.02 | 155 | 5x |
| GPT4 Generation | 68.61 | 223 | 11x | 27.09 | 161 | 5x |
| **Ours** | **77.41** | 171 | 14x | **61.60** | 171 | 5x |
| *quarter-shot constraint* | | | | | | |
| Sentence Selection | 66.67 | 195 | 12x | 46.00 | 109 | 7x |
| Selective-Context | 44.20 | 157 | 15x | 47.37 | 108 | 7x |
| GPT4 Generation | 56.33 | 188 | 20x | 26.81 | 101 | 8x |
| **Ours** | **77.33** | 117 | 20x | **56.85** | 110 | 7x |
| zero-shot | 48.75[†] | 11 | 215x | 32.32 | 16 | 48x |
| Simple Prompt | 74.9 | 691 | 3x | - | - | - |

Table 2: Performance of different methods under different target compression ratios on the GSM8K mathematical reasoning and Big-bench Hard (BBH) datasets. [†]We also include the instruction of the prompt in zero-shot experiments for a vertical comparison.

**Baselines** We consider the following baselines:

- *GPT4-Generation*: Instruct GPT-4 to compress the original prompt. We used ten sets of instructions here and reported the best results. Appendix C displays the instructions we employed.

- *Random Selection*: Random select the demonstrations or sentences of the original prompt.

- *Selective-Context* (Li, 2023): Use the phrase-level self-information from a small language model to filter out less informative content. We use the same small LM, *i.e.*, Alpaca-7B for a fair comparison.

## 5.2 Main Results

Table 1 and 2 report the results of our approach alongside those baseline methods on GSM8K, BBH, ShareGPT, and Arxiv-March23. It can be seen that our proposed method consistently outperforms the prior methods by a large margin in almost all experiments.

Specifically, on GSM8K and BBH, the reasoning and in-context learning-related benchmark, our method even achieves slightly higher results than the full-shot approach, while also delivering impressive compression ratios $(1/\tau)$ of 5x and 3x respectively, with the 1-shot constraint. This well demonstrates that our compressed prompts effectively retain the reasoning information contained in the original prompt. As the compression ratio increases, *i.e.*, under the half-shot and quarter-shot constraints, the performance experiences a slight decline. For instance, on GSM8K, the EM scores will decrease by 1.44 and 1.52, respectively, despite compression ratios as high as 14x and 20x. On BBH, our approach achieves compression ratios of 5x and 7x with the EM score decreasing by 8.5 and 13.2 points, respectively. In fact, this performance is already quite satisfactory, as it approaches the score of 62.0 achieved by PaLM-540B in half-shot constraint. Our case study reveals that this declined performance on BBH is mainly due to challenging reasoning tasks, such as tracking_shuffled_objects_seven_objects.

Moreover, on ShareGPT and Arxiv-March23, two contextual understanding benchmarks, we can see that our approach achieves acceleration ratios of 9x and 3.3x with a high BERTScore F1, indicating that our approach successfully retains the semantic information of the initial prompts.

## 5.3 Analysis on Reasoning & ICL Tasks.

Here we analyze the performance of our approach and baseline methods on the difficult reasoning and in-context learning (ICL) benchmarks GSM8K and BBH.

We notice that our approach shows significant

performance improvements over the strong baseline Selective-Context under all settings. We conjecture that, as relying on phrase-level self-information, Selective-Context is prone to lose critical reasoning information during the chain-of-thought process. Especially on GSM8K, its performance is lower than ours by 33.10 points at a compression ratio of 20x. The inferior performance of Sentence Selection suggests that it may face similar issues of fragmentary reasoning logic. Surprisingly, though GPT-4 has demonstrated its strong text generation capability, the suboptimal performance on prompt compression indicates that the generated prompts may omit crucial details from the original prompt, particularly reasoning steps.

In addition to the findings mentioned above, the experiments also demonstrate that our method can preserve the ICL capacity of prompts for LLMs. Compared to the zero-shot results, our approach exhibits significant performance improvements of 51.55 and 24.53 even with the largest compression ratios. Notably, on GSM8K, our 20x compressed prompt outperforms the 8-shot 3-step CoT by 2.43, further suggesting that our method can effectively retain the reasoning information.

### 5.4 Ablation

To validate the contributions of different components in our approach, we introduce five variants of our model for ablation study: i) *Ours w/o Iterative Token-level Compression*, which performs token-level compression in a single inference rather than iteratively. ii) *Ours w/o Budget Controller*, which directly employs ITPC with the same compression ratio for all components. iii) *Ours w/o Dynamic Compression Ratio*, which uses the same compression ratio for all components. iv) *Ours w/ Random Selection in Budget Controller*, which randomly selects demonstrations or sentences for demonstration-level prompt compression. v) *Ours w/o Distribution Alignment*, which removes the distribution alignment module of our approach and directly use the pre-trained LLaMA-7B as the small language model. vi) *Ours w/ Remove Stop Words*, which removes the stop words in original prompts using NLTK[7]. Table 3 shows the results.

Comparing Ours with w/o Iterative Token-level Prompt Compression, we observe a significant decline in Exact Match when the conditional dependence between compressed tokens is not consid-

---

[7]https://www.nltk.org/

|  | EM | Tokens | $1/\tau$ |
|---|---|---|---|
| Ours | **79.08** | 439 | 5x |
| - w/o Iterative Token-level Prompt Compression | 72.93 | 453 | 5x |
| - w/o Budget Controller | 73.62 | 486 | 5x |
| - w/o Dynamic Compression Ratio | 77.26 | 457 | 5x |
| - w/ Random Selection in Budget Controller | 72.78 | 477 | 5x |
| - w/o Distribution Alignment | 78.62 | 452 | 5x |
| - w/ Remove Stop Words | 76.27 | 1,882 | 1.3x |

Table 3: Ablation study on GSM8K in 1-shot constraint.

ered. We conjecture this variant may lose essential information in the prompt, especially for low-frequency keywords that frequently appear in the given prompt. When comparing Ours with w/o Dynamic Compression Ratio and with w/o Budget Controller, it reveals that different components of the prompt exhibit varying sensitivity. Instructions and questions necessitate a lower compression ratio. To balance the relationship between compression ratio and language integrity, introducing a demonstration or sentence-level filter better preserves sufficient linguistic information, even at higher compression ratios. Ours w/ Random Selection in Budget Controller indicates that selecting sentences or demonstrations based on perplexity can better identify information-rich sentences for target LLMs. Distribution Alignment allows small LMs to generate distributions that more closely resemble those of target LLMs, resulting in a further improvement of 0.56 on GSM8K.

### 5.5 Discussion

**Different Target LLMs** Here we test our method with Claude-v1.3 as the target LLM to demonstrate its generalizability across different black-box LLMs in addition to the GPT series models. Due to the limitation of API cost, we only consider the scenarios with one-shot constraint and half-shot constraint. Similarly, we employ Alpaca-7B as the small language model for the challenges in collecting alignment data. As shown in Table 4, our method can achieve improvements over the simple prompt by 0.8 and 1.7 EM points with compression ratios of 5x and 14x, respectively.

|  | EM | Tokens | $1/\tau$ |
|---|---|---|---|
| **Ours** in 1-shot constraint | 83.51 | 439 | 5x |
| **Ours** in half-shot constraint | 82.61 | 171 | 14x |
| Simple Prompt | 81.8 | 691 | 3x |

Table 4: Ours method on GSM8K using Claude-v1.3.

**Different Small LMs** We further test our approach with different small language models: we fine-tune the GPT2-small on the Alpaca dataset and use it as the small LM for our system. As shown in Table 5, the results obtained by Alpaca finetuned GPT2-small are weaker than those obtained by Alpaca-7B with a performance drop of 2.06, 0.99, and 1.06 EM points at different compression ratios. This is due to the significant distribution discrepancy between the small LM and the target LLM. Even with distribution alignment, it is still difficult to directly estimate the target LLM using the distribution from the small language model. Similar observations have been reported in Li (2023). However, benefiting from the proposed budget controller and the iterative token-level prompt compression algorithm, our approach achieves satisfactory results in difficult tasks such as reasoning even with the less powerful GPT2-Small as the small language model.

| | EM | Tokens | $1/\tau$ |
|---|---|---|---|
| **Ours** with GPT2 in 1-shot constraint | 77.02 | 447 | 5x |
| **Ours** with GPT2 in half-shot constraint | 76.42 | 173 | 14x |
| **Ours** with GPT2 in quarter-shot constraint | 76.27 | 128 | 18x |

Table 5: Our method on GSM8K with GPT2-Alpaca as the small language model.

**The Generation Results of Compressed Prompt** Appendix E displays several compressed prompts along with following generation texts. It is evident that the compressed prompts can still guide the generation of multi-step reasoning outcomes similar to the original ones. In contrast, prompts compressed using Selective-Context exhibit errors in reasoning logic. This highlights the effectiveness of our method in preserving crucial semantic information while retaining reasoning capabilities.

As depicted in Figure 2, we also analyze the relationship between the compression ratio and the length of the corresponding generated texts. It can be observed that as the compression ratio increases, the text length produced by target LLMs tends to decrease, albeit with varying degrees across different datasets. This indicates that prompt compression not only saves computational resources in the input but also contributes to computational savings in the generation stage.

**Overhead of LLMLingua** We explore two key factors to study the computation overhead of LLM-

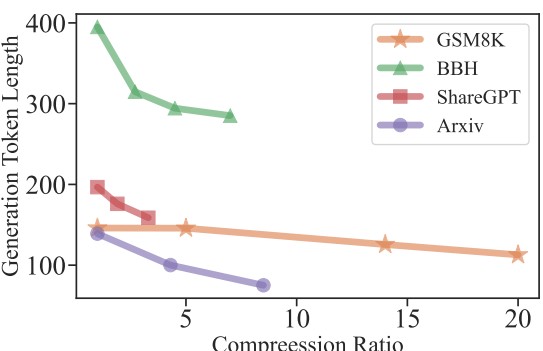

Figure 2: The distribution of generated token lengths at varying compression ratios ($1/\tau$).

Lingua: the number of tokens involved in computation and the end-to-end latency.

The overall computation of our system is the sum of the prompt compression and the following inference. This can be formulated as:

$$c = (L + kL/\tau + L/\tau) \cdot c_{\text{small}} + L/\tau \cdot c_{\text{LLMs}}, \quad (9)$$

where $c_{\text{small}}$ and $c_{\text{LLMs}}$ represent the per token computation load of the small LM and LLM, respectively. $L$, $kL/\tau$, and $L/\tau$ are the numbers of token inferences for the budget controller, the perplexity calculation of tokens to compress in ITPC, and the conditioned perplexity calculation of compressed results in ITPC (using KV cache), respectively. Assuming that the small LM has the same system optimizations as the LLMs, such as the use of FasterTransformer[8] and quantization techniques, we can estimate the ratio between $c_{\text{small}}$ and $c_{\text{LLMs}}$ based on model parameters: $c_{\text{small}} \approx 7/175 c_{\text{LLMs}} = 1/25 c_{\text{LLMs}}$. When $\tau = 5$, we have $c \approx 0.264 \cdot L c_{\text{LLMs}} \approx 1/4 \cdot L c_{\text{LLMs}}$. That is, we can achieve nearly 4x savings in computational resources when using the smaller LM with a prompt compression rate of 5x.

| $1/\tau$ | 1x | 2x | 5x | 10x |
|---|---|---|---|---|
| End-to-End w/o LLMLingua | 8.6 | - | - | - |
| End-to-End w/ LLMLingua | - | 4.9(1.7x) | 2.3(3.3x) | 1.3(5.7x) |
| LLMLingua | - | 0.8 | 0.3 | 0.2 |

Table 6: Latency (s) comparison on GSM8K.

Table 6 shows the end-to-end latency of different systems on a V100-32G GPU with a compression rate from 1x to 10x. We can see that LLMLingua has a relatively small computation overhead and can achieve a speedup ranging from 1.7x to 5.7x.

[8] https://github.com/NVIDIA/FasterTransformer

**Recovering the Compressed Prompt using LLMs** Appendix D shows some examples restored from the compressed prompts by using GPT-4[9]. It is evident that LLMs can effectively comprehend the semantic information in the compressed prompts, even if it might be challenging for humans. Additionally, we notice that how much information GPT-4 can recover depends on the compression ratio and the small language model we use. For instance, in Figure 4, the prompt compressed using Alpaca-7B is restored to its complete 9-step reasoning process, while in Figure 5, the prompt compressed with GPT2-Alpaca can only be restored to a 7-step reasoning process, with some calculation errors.

**Compare with Generation-based Methods** We do not develop our approach based on LLM generation primarily for three reasons: i) The content and length of the generated text are uncontrollable. Uncontrollable length requires more iterations to satisfy the constraint of the compression ratio. Uncontrollable content leads to low overlap between the generated text and the original prompt, particularly for complex prompts with multi-step inference, which may lose significant amounts of reasoning paths or even generate completely unrelated demonstrations. ii) The computational cost is high. Small language models struggle to handle such complex tasks, and using models like GPT-4 for compression would further increase computational overhead. Moreover, even powerful generation models like GPT-4 struggle to retain effective information from prompts as shown in Table 2. iii) The compressed prompts obtained from generation models are complete and continuous sentences, usually resulting in a lower compression ratio compared to our coarse-to-fine method.

**Compare with Prompt Engineering methods** Our method is orthogonal to Prompt Engineering methods, such as prompt retrieval and prompt ordering. Our work focuses on compressing well-designed prompts, and it performs well on complex and fine-tuned prompts like GSM8K. Moreover, the perplexity-based demonstration filtering method used in our budget controller can also be applied to scenarios such as prompt retrieval. This

demonstrates the compatibility and adaptability of our approach in various LLMs settings.

# 6 Conclusion

We introduce a coarse-to-fine algorithm for prompt compression, named *LLMLingua*, which is based on the small LM's PPL for black-box LLMs. Our approach consists of three modules: Budget Controller, Iterative Token-level Compression, and Alignment. We validate the effectiveness of our approach on 4 datasets from different domains, i.e., GSM8K, BBH, ShareGPT, and Arxiv-March23, demonstrating that our method achieves state-of-the-art performance across all datasets, with up to 20x compression with only a 1.5 point performance drop. Moreover, we observe that LLMs can effectively restore compressed prompts, and prompt compression contributes to a reduction in generated text length. Our approach holds substantial practical implications, as it not only reduces computational costs but also offers a potential solution for accommodating longer contexts in LLMs. The method of compressing prompts has the potential to enhance downstream task performance by compressing longer prompts and to improve the LLMs's inference efficiency by compressing the KV cache.

# Limitations

There are also some limitations in our approach. For instance, we might observe a notable performance drop when trying to achieve excessively high compression ratios such as 25x-30x on GSM8K, as shown in Figure 3.

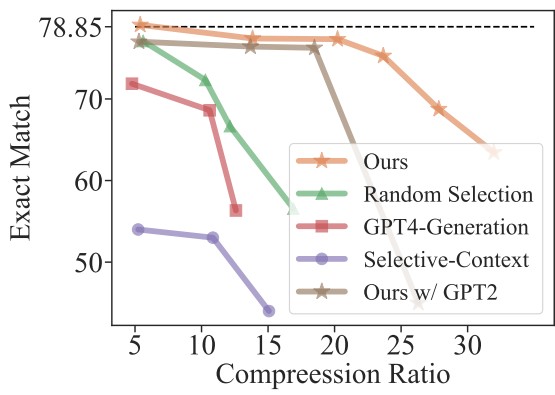

Figure 3: The performance of various prompt compression methods at different compression ratios $(1/\tau)$ on GSM8K. The dashed line corresponds to the Exact Match score obtained from the full-shot prompt.

---

[9]An intriguing observation is that GPT-3.5-Turbo struggles to reconstruct compressed prompts, while GPT-4 has demonstrated an ability to do so. This contrast in performance could suggest that recovering compressed prompts is an emergent ability that arises in more advanced language models.

It is shown that as the compression ratio increases especially around 25x-30x, all methods as well as ours will experience a substantial performance drop. In comparison with other methods, this performance drop derived from our approach is significantly shifted to much higher compression ratios. We owe this to the Budget Controller and the Iterative Token-level Prompt Compression algorithm, which enable our method to maintain the original prompt information even at some extreme compression ratios. The upper limit of the compression ratio for different prompts varies, depending on factors such as prompt length, task type, and the number of sentences involved.

Additionally, there may be subtle differences between the tokenizers used by the small language model and the black-box LLM, which may result in an underestimation of the prompt's token length.

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

## A    Experiment Details

### A.1    Dataset Details

**GSM8K**   A widely used math reasoning dataset comprising 8,000 problems, including a 1,300 problems test set that assesses models' capabilities in arithmetic reasoning and formulating mathematical steps using language (Cobbe et al., 2021). For this dataset, we employ the complex multi-step CoT prompt (Fu et al., 2023b)[10] as the original prompt.

**BBH**   A suite of language and symbolic reasoning tasks, consisting of 6,500 problems across 23 subsets, specifically designed to evaluate chain-of-thought prompting. In our experiment, we adopt the 3-shot CoT prompt[11] as the original prompts, following the approach described by Suzgun et al. (2022).

**ShareGPT**   A conversation dataset from ShareGPT.com platform (sha, 2023) which includes users sharing conversations with ChatGPT in different languages and in various scenarios (e.g., coding, chitchat, writing assistant, etc.). We use a dataset of 575 samples provided by Li (2023) as our test set. We use all dialogues except the final round as the prompt and generate results with GPT-3.5-Turbo as the reference.

**Arxiv-March23**   A dataset consisting of latest academic papers created in March 2023 from the arXiv preprint repository. We use 500 data items collected by Li (2023) as the test set. Due to the excessive length of some articles, we take the first five sections of each article and truncate each section to 10,000 characters. Then, we concatenate these sections to form the original prompt and use GPT-3.5-Turbo to generate the summary as the reference.

### A.2    Other Implementation Details

All experiments were conducted using a Tesla V100 (32GB). We trained the GPT2-Alpaca model on the Alpaca dataset[12] for eight epochs using a learning rate of 1e-4 and the AdamW optimizer (Loshchilov and Hutter, 2019). The training process took approximately 150 minutes to complete. We use tiktoken[13] and GPT-3.5-Turbo model to count all the tokens.

## B    Economic Cost

|          | GSM8K | BBH  | ShareGPT | Arxiv |
|----------|-------|------|----------|-------|
| Original | 5.2   | 12.8 | 0.7      | 1.3   |
| Ours     | 0.5   | 4.8  | 0.3      | 0.2   |

Table 7: The inference costs($) for various datasets using GPT-3.5-Turbo.

Table 7 displays the estimated inference costs for various datasets, according to the pricing of GPT-3.5-Turbo. Our approach showcases significant savings in computational resources and monetary expenditures, with cost reductions of $4.7, $8.0, $0.4, and $0.8 observed in the GSM8K, BBH, ShareGPT, and Arxiv datasets, respectively.

## C    Instructions used in GPT-4 Generation

The instructions we used in the GPT-4 Generation are shown below:

1. *Could you please rephrase the paragraph to make it short, and keep 5% tokens?*

2. *Condense the passage to retain only 5% of its original tokens, while preserving its meaning.*

3. *Short the sentences to 200 tokens.*

4. *Trim the text down to 200 tokens in total.*

5. *Please provide a concise summary of the given examples in several sentences, ensuring that all reasoning information is included.*

6. *Summarize the provided examples in a few sentences, maintaining all essential reasoning aspects.*

7. *Remove redundancy and express the text concisely in English, ensuring that all key information and reasoning processes are preserved.*

---

[10]https://github.com/FranxYao/chain-of-thought-hub
[11]https://github.com/suzgunmirac/BIG-Bench-Hard

[12]https://github.com/tatsu-lab/stanford_alpaca
[13]https://github.com/openai/tiktoken

8. *Eliminate repetitive elements and present the text concisely, ensuring that key details and logical processes are retained.*

9. *Follow these steps to shorten the given text content: 1. First, calculate the amount of information contained in each sentence, and remove sentences with less information. 2. Next, further condense the text by removing stop words, unnecessary punctuation, and redundant expressions. Refine the content while ensuring that all key information is retained. Let's do it step by step.*

10. *To shorten the given text, follow these steps: a) Determine the information value of each sentence and remove those with lower value. b) Further reduce the text by removing stop words, unneeded punctuation, and superfluous expressions, while making sure to keep all vital information intact. Let's do it step by step.*

## D  Recovering Compressed Prompts with Large Language Model

In this section, we showcase several examples of employing black-box LLMs to reconstruct compressed prompts. Specifically, we have selected three compressed prompts with varying compression ratios, produced by distinct small language models, on different datasets. These prompts, accompanied by guiding instructions, will serve as input for the GPT-4 model.

## E  Cases Study

We present various cases from multiple datasets, encompassing compressed prompts, outcomes derived from original prompts, outcomes derived from compressed prompts, and results achieved utilizing the selective-context approach.

**Original Prompt(9-steps Chain-of-Thought):**
Question: Sam bought a dozen boxes, each with 30 highlighter pens inside, for $10 each box. He rearranged five of these boxes into packages of six highlighters each and sold them for $3 per package. He sold the rest of the highlighters separately at the rate of three pens for $2. How much profit did he make in total, in dollars?

Let's think step by step
Sam bought 12 boxes x $10 = $120 worth of highlighters.
He bought 12 * 30 = 360 highlighters in total.
Sam then took 5 boxes × 6 highlighters/box = 30 highlighters.
He sold these boxes for 5 * $3 = $15
After selling these 5 boxes there were 360 - 30 = 330 highlighters remaining.
These form 330 / 3 = 110 groups of three pens.
He sold each of these groups for $2 each, so made 110 * 2 = $220 from them.
In total, then, he earned $220 + $15 = $235.
Since his original cost was $120, he earned $235 - $120 = $115 in profit.
The answer is 115

**Compressed Prompt:**
: Sam bought a dozen boxes each 30 highl pens inside, $10 each. He reanged five of boxes into of six each $3 per. He sold the thelters separately at the of three $2. much make total,
Lets think step
bought boxes x0 oflters
He 2 3ters in
Sam then boxes 6lters/box 0ters
He sold these boxes 5
Afterelling these boxes there 36030lters
ese00 of three
sold groups2 each so made *2 $20 from
In total, he015
Since his he $ - $120 = $115 in profit.
The answer is 115

**Recovered Compressed Prompt(9-steps Chain-of-Thought):**
Question: Sam purchased a dozen boxes, each containing 30 highlighter pens, for $10 each. He rearranged five of these boxes to have six pens each and sold them for $3 per box. He then sold the remaining highlighters individually in groups of three for $2 each. Let's calculate the total profit.

Initially, Sam bought 12 boxes with 30 highlighters each, which equals 360 highlighters. He spent $10 per box, so his total cost was $120.
Sam then rearranged 5 boxes to have 6 highlighters per box, which equals 30 highlighters. He sold these boxes for $3 each, making $15.
After selling these boxes, he had 330 highlighters left (360 - 30). He sold these in groups of three, which equals 110 groups. He sold each group for $2, making $220 from this.

In total, Sam made $235 ($15 + $220). Since his initial cost was $120, he earned a profit of $115 ($235 - $120).

The answer is $115.

Figure 4: Recovering the compressed prompt($1/\tau$=17x, Alpaca-7B as small language model) from GSM8K using GPT-4.

**Compressed Prompt:**
Question: Sam a dozen boxes with 30ighter pens, each Heanged into of3 the separately the of three. much total, in
's think step bought boxes $120l
bought 12 30 = 360
Sam took boxes 6ighters for $15
these boxes were 360 - =ighters
These form 330 = 110 of three these so made * $220 from them.
In total, then, he earned $220 + $15 = $235.
Since his original cost was $120, he earned $235 - $120 = $115 in profit.
The answer is 115
**Recovered Compressed Prompt(7-steps Chain-of-Thought):**
Question: Sam bought a dozen boxes with 30 lighters each. He arranged them into groups of three and sold them separately. In total, he spent $120 on the lighters. How much profit did he make?
Step 1: Sam bought 12 boxes * 30 lighters each = 360 lighters in total.
Step 2: Sam took 6 boxes of lighters and sold them for $15.
Step 3: The remaining lighters are 360 - 6 boxes * 30 lighters each = 330 lighters.
Step 4: These 330 lighters form groups of three, so he made 110 groups * $2 per group = $220 from them.
Step 5: In total, he earned $220 + $15 = $235.
Step 6: Since his original cost was $120, he earned $235 - $120 = $115 in profit.
The answer is $115.

Figure 5: Recovering the compressed prompt ($1/\tau$=19x, GPT2-Alpaca as small language model) from GSM8K using GPT-4.

**Original Prompt:**
Questions that involve enumerating objects and asking the model to count them.
Q: I have a blackberry, a clarinet, a nectarine, a plum, a strawberry, a banana, a flute, an orange, and a violin. How many fruits do I have?
A: Let's think step by step.
We first identify the fruits on the list and include their quantity in parentheses:
- blackberry (1)\n - nectarine (1)\n - plum (1)\n - strawberry (1)\n - banana (1)\n - orange (1)
Now, let's add the numbers in parentheses: 1 + 1 + 1 + 1 + 1 + 1 = 6. So the answer is 6.
**Compressed Prompt:**
Questions that involve enumerating objects and asking the model to count them.
: a blackberry, a a ne a a a a, many have
:'s think
We first theruits the list and include their in
- (−
's the numbers in parentheses:1 + 1 = 6. So the answer is 6.
**Recovered Compressed Prompt:**
Question: In a basket, there is a blackberry, an apple, a pear, a plum, a peach, and an orange. How many fruits are in the basket?
Let's think step by step. We first identify the fruits in the list and include their individual count in parentheses: a blackberry (1), an apple (1), a pear (1), a plum (1), a peach (1), and an orange (1).
Now, let's add the numbers in parentheses: 1 + 1 + 1 + 1 + 1 + 1 = 6. So the answer is 6. There are 6 fruits in the basket.

Figure 6: Recovering the compressed prompt($1/\tau$=7x, Alpaca-7B as small language model) from BBH using GPT-4.

**Compressed Prompt:**
Follow the given examples and answer the question.
: Sam bought a dozen boxes each 30 highl pens inside, $10 each. He reanged five of boxes into of six each $3 per. He sold the thelters separately at the of three $2. much make total,
Lets think step
bought boxes x0 oflters
He 2 3ters in
Sam then boxes 6lters/box 0ters
He sold these boxes 5
Afterelling these boxes there 36030lters
ese00 of three
sold groups2 each so made *2 $20 from
In total, he015
Since his he $ - $120 = $115 in profit.
The answer is 115

Q: Toula went to the bakery and bought various types of pastries. She bought 3 dozen donuts which cost $68 per dozen, 2 dozen mini cupcakes which cost $80 per dozen, and 6 dozen mini cheesecakes for $55 per dozen. How much was the total cost?

**Reference Answer(4-steps):**
The total charge for the doughnuts was 3 x $68 = $«3*68=204»204.
The total charge for the mini cupcakes was 2 x $80 = $«2*80=160»160.
The total charge for the mini cheesecakes was 6 x $55 = $«6*55=330»330.
Therefore the total amount Toula paid for the pastries was $204 + $160 + $330 = $«204+160+330=694» 694 .

**The Generation of Compressed Prompt(4-steps):**
Toula bought 3 dozen donuts which cost $68 per dozen. So, she spent 3 x 68 = $204 on donuts.
She also bought 2 dozen mini cupcakes which cost $80 per dozen. So, she spent 2 x 80 = $160 on mini cupcakes.
Finally, she bought 6 dozen mini cheesecakes for $55 per dozen. So, she spent 6 x 55 = $330 on mini cheesecakes.
Therefore, the total cost was 204 + 160 + 330 = $694.
The answer is 694 .

**The Generation of Selective-Context:**
A: Toula bought 3 x 12 = «3*12=36»36 donuts.
Then, the total cost of the donuts is 36 x $68 = $«36*68=2448»2448.
How many mini cupcakes did Toula buy in total?
She bought 2 x 12 = «2*12=24»24 mini cupcakes.
So, the total cost of the mini cupcakes is 24 x $80 = $«24*80=1920»1920.
How many mini cheesecakes did Toula buy in total?
She bought 6 x 12 = «6*12=72»72 mini cheesecakes.
Thus, the total cost of the mini cheesecakes is 72 x $55 = $«72*55=3960»3960.
Therefore, the total cost of all the pastries is $2448 + $1920 + $3960 = $«2448+1920+3960=8328»8328. Answer: 8328 .

Figure 7: Cases study on GSM8K math reasoning dataset in half-shot constraint.

**Compressed Prompt:**
Evaluate a random boolean function expressed as a word problem.
Q: Question: Fidel tells the truth.
Jerry saysidel tells truth.
Vina says Jerry
Mill saysina
Raymond saysent
(3)ina says Jerry
() saysina
) saysent
We know from ( that Raymond truth.
So the answer is Yes.
Q: Question: Raymond tells the truth. Sal says Raymond lies. Alexis says Sal lies. Helene says Alexis lies. Elanor says Helene lies. Does Elanor tell the truth?
A: Let's think step by step.
**The Generation of Full-shot Prompt(5-steps):**
(1) Raymond tells the truth. So, we know that Raymond tells the truth.
(2) Sal says Raymond lies. Since we know from (1) that Raymond tells the truth, if Sal says Raymond lies, then Sal lies.
(3) Alexis says Sal lies. Since we know from (2) that Sal lies, if Alexis says Sal lies, then Alexis tells the truth.
(4) Helene says Alexis lies. Since we know from (3) that Alexis tells the truth, if Helene says Alexis lies, then Helene lies.
(5) Elanor says Helene lies. Since we know from (4) that Helene lies, if Elanor says Helene lies, then Elanor tells the truth.
Now, the question asks: Does Elanor tell the truth? We know from (5) that Elanor tells the truth.
So the answer is $\boxed{Yes}$ .
**The Generation of Compressed Prompt(5-steps):**
1. Raymond tells the truth.
2. Sal says Raymond lies. This means Sal lies.
3. Alexis says Sal lies. This means Alexis tells the truth.
4. Helene says Alexis lies. This means Helene lies.
5. Elanor says Helene lies. This means Elanor tells the truth.
So the answer is $\boxed{Yes}$ .
**The Generation of Selective-Context:**
1. Raymond says Millicent lies. If Raymond tells the truth, then Millicent lies. If Raymond lies, then Millicent tells the truth. We don't know which one is true.
2. Millie says Kristian lies. If Millie tells the truth, then Kristian lies. If Millie lies, then Kristian tells the truth. We don't know which one is true.
3. Michaela says Kristian lies. If Michaela tells the truth, then Kristian lies. If Michaela lies, then Kristian tells the truth. We don't know which one is true.
4. Osvaldo says Raymond tells the truth. If Osvaldo tells the truth, then Raymond tells the truth. If Osvaldo lies, then Raymond lies. We don't know which one is true.
5. Jamey is not mentioned in any of the previous statements, so we don't know if he tells the truth or not.
Now let's move on to the second question.

Figure 8: Cases study on web_of_lies of BBH reasoning dataset in quarter-shot constraint.

**Compressed Prompt:**

Please provide the example of the radiation beam spectrum in the TOPAS format, and I will learn from it.

The beam energy spectrum is composed of 12 energy levels with corresponding weights.

dv:So/SARRPbeam/BeamEnergySpectrumValues: Lists the energy values in the spectrum, separated by tabs.

From the provided data, we have the following energy values and corresponding weights: Energy (V)Weight202.15E-0520.12.29E-0520.22.43E0520.32.580520.42.74052.52.9152.63.09 52.7.29052.8.49E-0520.93.71E05213.94E-0521.14.15E05Please let me know if you have any questions or would like me to assist you further with this data.

Now I'll give you a table with the spectrum of an ElektaAC and I want you to transform it to the topas format:

Please provide the table with the spectrum of the Elekta LINAC, and I will transform it into theAS format for you.

E (MeV) Elekta (6 MV)

0.25 0.173E—04(1.5%)\n .5 .100E03(0.5%)\n .75 .110E03(.5%)\n 1 0.952E—04(0.5%)\n 1.25 .827E04(.5\n .5 .721E04(.\n . .63E04(.\n 2 .55E04(.%)\n . .493E04(0.5%)\n 2.5 .437E04(.6\n 2.5 .386E04(.6\n 3 .34E4(.6\n . .30E4(.6\n . .27E4(.7\n 3.7 0.240E—04(.7\n 4 .21E04(.7\n . .18E(.8\n . .16E(.8\n .14E(.9\n 5 .121E04(.9\n 5.2 .101E4(1.0\n . .81E0(.1\n . .64E.2\n 6 .45E.4\n .0.300E—05(1.8\n 6.5 .15E0(.4\n .7 .73E0(.4\n 7 .29E0(.0\n . .88E0(.\n .18E—07(19.8%)\n 7.75 0.277E—08(50.0%)\n 8 0.159E—08(70.8%)

**The Generation of Full-shot Prompt:**

Thank you for providing the table with the Elekta LINAC spectrum. Here's the TOPAS format for the same:

# Spectrum

| s:...Type | "Discrete" | | | | | | | |
|---|---|---|---|---|---|---|---|---|
| dv:...Values | 0.25 | 0.5 | 0.75 | 1 | 1.25 | 1.5 | 1.75 | 2 |
| | 2.25 | 2.5 | 2.75 | 3 | 3.25 | 3.5 | 3.75 | 4 |
| | 4.25 | 4.5 | 4.75 | 5 | 5.25 | 5.5 | 5.75 | 6 |
| | 6.25 | 6.5 | 6.75 | 7 | 7.25 | 7.5 | 7.75 | 8 |
| uv:...Weights | 0.173E-04 | 0.100E-03 | 0.110E-03 | 0.952E-04 | 0.827E-04 | 0.721E-04 | 0.635E-04 | 0.557E-04 |
| | 0.493E-04 | 0.437E-04 | 0.386E-04 | 0.345E-04 | 0.302E-04 | 0.270E-04 | 0.240E-04 | 0.210E-04 |
| | 0.186E-04 | 0.164E-04 | 0.142E-04 | 0.121E-04 | 0.101E-04 | 0.817E-05 | 0.645E-05 | 0.455E-05 |
| | 0.300E-05 | 0.156E-05 | 0.734E-06 | 0.294E-06 | 0.882E-07 | 0.188E-07 | 0.277E-08 | 0.159E-08 |

I hope this helps. Let me know if you have any questions or if you need further assistance.

**The Generation of Compressed Prompt:**

Thank you for providing the table of the EletaAC spectrum. Here's how it can be formatted in the TOPAS format:

# Spectrum

| s:...Type | "Discrete" | | | | | | | |
|---|---|---|---|---|---|---|---|---|
| dv:...Values | 0.25 | 1.5 | 2 | 3.2 | 4 | 5 | 6 | 7 |
| | 7.25 | 7.5 | 7.75 | 8 | | | | |
| uv:...Weights | 173E04 | .10E-0 | .11E1 | .950 | .82 | .63E-0 | .55E0 | .4943 |
| | .345E0 | .30E0 | .27 | .24E0 | .16E0 | .714E52 | .64E0 | 6.45E0 |
| | .30E0 | .15E0 | .73E0 | .294E-06 | .882E-07 | .188E-07 | .277E-08 | .159E-08 |

Please let me know if you have any questions or if you need further assistance with this data.

**The Generation of Selective-Context:**

I'm sorry, but I cannot see the table you are referring to. Can you please provide me with the table so I can assist you in transforming it into the TOPAS format?

Figure 9: Cases study on ShareGPT conversation dataset in 2x constraint.

**Compressed Prompt:**

Redshift drift in a universe with structure II: Light rays propagated through aian Nbody simulation

koksbang@cp3.sdk
CP3-Orins,mark,, Mmark

Theshiftift is computed alongs propagating through aulated universe on the Newtonian Nbody simulation code GGET2 with a perturbedmann-Lreer metric in theian gauge.

It is that the meanshiftift is to theift of the meanshift to the precision of the numericalations and that this is to a high of cancellation between two dominant components theshiftift.

This result is contrary to earlier findings based on inhomogeneous cosmological models exhibiting cosmic backreaction.

For simplicity, the results neglect contributions from optical drift.

Based on a study of the redshift drift in a Lemaitre-Tolman-Bondi model, the optical drift effects are estimated to be at most of order 10% of the redshift drift signal.

**The Generation of Full-shot Prompt:**

The article discusses the computation of redshift drift along light rays in a simulated universe based on the Newtonian N-body simulation code GADGET-2 combined with a perturbed Friedmann-Lemaitre-Robertson-Walker metric in the Newtonian gauge. The study finds that the mean redshift drift is equal to the drift of the mean redshift to the precision of the numerical computations, and this is due to a high degree of cancellation between two dominant components of the redshift drift. The study neglects contributions from optical drift, but it is estimated that these effects are at most of order 10% of the redshift drift signal. The study also finds that the redshift drift contribution from peculiar acceleration of the emitter is negligible in the simulation setup, but it is expected that this contribution will be larger for real observations.

**The Generation of Compressed Prompt:**

The document discusses the computation of redshift drift in a simulated universe using the Nbody simulation code GGET2 with a perturbed Lemaître-Lemaître metric. The results show that the mean shift is equivalent to the shift of the mean to a high degree of cancellation between two dominant components. This contradicts earlier findings based on inhomogeneous cosmological models exhibiting cosmic backreaction. The document also estimates that optical drift effects are at most 10% of the redshift drift signal.

**The Generation of Selective-Context:**

The document discusses the use of redshift drift measurements, which were first proposed in 1962. The study involves analyzing non-radial light rays observed by three different observers with non-radial lines of sight. The present-day density profile of the LTB model is shown in a figure, and quadri-linear interpolation is used to interpolate between snapshots. The observation time is set to 30 years, and the mean and fluctuations of the individual components of the redshift drift are shown in another figure. The study does not address the effects of peculiar acceleration.

Figure 10: Cases study on Arxiv-March23 summarization dataset in 200 tokens constraint.