# OpenReview forum: "LLMLingua: Compressing Prompts for Accelerated Inference of Large Language Models"
_EMNLP/2023/Conference — EMNLP 2023 Main_

### Official Review · Reviewer_ZEXc · 2023-08-01

**Soundness:** 4

**Excitement:**

3: Ambivalent: It has merits (e.g., it reports state-of-the-art results, the idea is nice), but there are key weaknesses (e.g., it describes incremental work), and it can significantly benefit from another round of revision. However, I won't object to accepting it if my co-reviewers champion it.

**Paper Topic And Main Contributions:**

This paper presents a novel approach for LLM prompt compression. The authors introduce a two-step method using an auxiliary (small) model : they first drop demonstrations that have the lowest perplexity, and then use a hierarchical selection approach to extract the more relevant tokens in the whole prompt. They show that their approach significantly outperforms previous prompt compression methods across various compression constraints on several datasets.

**Questions For The Authors:**

1 - L348-349, you mention using a "top p" of 1 and a "decoding temperature" of 0 - which if I correctly understand "decoding temperature" should not be possible - for generation. Do you mean that you are using greedy (or argmax) decoding?

2 - In Table 1, is there a specific reason why you did not include full-shot and zero-shot results?

3 - L359-360, you describe your GPT4-Generation baseline as "Instruct GPT4 to compress the original prompt". You then point out (L422-423) that "the generated prompts may omit crucial details from the original prompt, particularly reasoning steps". A naive way to try and fix this would be to tune the prompt of GPT4 so that it avoids omitting such details and reasoning steps, which may lead to a stronger baseline. What prompt(s) did you use for the compression instruction? Did you tune it/them to make the baseline stronger?

4 - How much compute is required by the two-step method to produce the compressed prompt? How does that affect the latency of the overall system (from original prompt to answer)? If my understanding of the paper is correct, you need roughly #tokens//100 + #demonstrations forward passes in the auxiliary model?

**Reasons To Accept:**

The method presented in this paper is novel and more subtle than previous approaches. The authors conducted convincing experiments to support their method. The impact of their findings may be of interest for the community as it seems to reduce the cost of LLM prompt-based generation.

**Reasons To Reject:**

The presented method relies on many different steps, and uses a lot of auxiliary computation, which is not discussed in the paper. *Important comparison points are missing in Table 1* => After rebuttal, this concern has been addressed, and the Table 1 is convincing in my opinion.

**Reproducibility:**

4: Could mostly reproduce the results, but there may be some variation because of sample variance or minor variations in their interpretation of the protocol or method.

**Reviewer Confidence:**

4: Quite sure. I tried to check the important points carefully. It's unlikely, though conceivable, that I missed something that should affect my ratings.

**Typos Grammar Style And Presentation Improvements:**

### Typos
L148
token merging( => token merging (

Algorithm 2
budget controlloer => budget controller

L347 (& others)
Cluade API => Claude API

### Style
L191-192: The smaller value of 𝜏, the lower the inference cost, and the more we prefer => strange phrasing in my opinion
"A smaller value of 𝜏 implies a lower inference cost, which is preferable." would be easier to read?

---

> ### Author Rebuttal · Authors · 2023-08-28
>
> Thank you for pointing out the potential confusion and providing insightful feedbacks.
>
> **1. “The presented method relies on many different steps, and uses a lot of auxiliary computation, which is not discussed in the paper”**
>
> Our empirical observation reveals that prompt compression requires balancing information at different granularities. To this end, we design our approach as a pipeline method involving several steps. The necessity and effectiveness of each step are validated in Section 5.4 and 5.5.
>
> The computation derived from our system is much less than that of LLMs with the original prompt. Specifically, the computation load of our system is the sum of the computation required for prompt compression and the computation needed for inference on the compressed prompt, which can be formulated as:
>
> $c = (L + k \cdot \frac{L}{\tau} + \frac{L}{\tau}) c_S + \frac{L}{\tau} c_L$,
>
> where $c_S$ and $c_L$ represent the per token computation load of small LMs and LLMs, $L$ denotes the length of the original prompt, $k$ is the granular control coefficient, and $\tau$ is the compression ratio. $L$, $ k \cdot \frac{L}{\tau} $, and $\frac{L}{\tau} $ are the numbers of token inferences for the budget controller, the first perplexity calculation and condition perplexity calculation (using KV cache to reduce redundant computations) in iterative token-level prompt compression, respectively. Given that the number of parameters in GPT-3.5-turbo (175B) is 25 times that of LLaMA-7B, we can set $c_S = \frac{1}{25} c_L$. Assuming $k=2$ (as used in the paper) and $\tau=5$, we obtain $c = 0.264 \cdot Lc_L \approx 1/4 \cdot Lc_L$, where $Lc_L$ represents the computation required when LLMs process the original prompt.
> Taking into account the cost and price of the API, as well as the reusability of prompts in the dataset, the cost of the compressed prompt system would be further reduced, such that $c < 0.0264 \cdot Lc_L$. We will add this analysis in the final version.
>
> **2. “Important comparison points are missing in Table 1.”**
>
> Since there are no further details provided, we can assume that this comment aligns with the second item in the 'questions' section. For more information, please refer to the fourth response.
>
> **3. “top p of 1 and a decoding temperature of 0… do you mean that you are using greedy decoding?”**
>
> Yes, during the LM decoding process, a temperature of 0 is equivalent to greedy decoding. To report stable and reproducible results, we used greedy decoding in all our experiments.
>
> **4. “In Table 1, why you did not include full-shot and zero-shot results”**
>
> Table 2 is about few-shot QA tasks and thus includes the setting of full-shot and zero-shot. However, results shown in Tabel 1 are derived from conversation and summarization tasks. In this context, references used for evaluation are generated from the original prompts, and thus can be regarded as the full-shot prompts (detailed description can be found in Appendix A.1).
>
> Since the zero-shot setting is not applicable to the tasks in Table 1, here we conduct experiment with the last sentence as an extremely short prompt for comparison. It can be seen that although the response’s topic aligns with the ground truth, the lack of detail and context results in relatively poor n-gram metrics and BERT Score.
>
> | ShareGPT | BLEU | Rouge1 | Rouge2 | RougeL | BS F1 | Tokens | 1/$\tau$ |
> |- | - | - | - | - | - | - | - |
> | **Ours** 3x constraint | 19.55 | 40.81 | 22.68 | 30.98 | 87.70 | 177 | 3.3x |
> | Last Sentence | 11.81 | 30.02 | 13.33 | 21.44 | 82.54 | 28 | 21.1x |
>
> | Arxiv-March23 | BLEU | Rouge1 | Rouge2 | RougeL | BS F1 | Tokens | 1/$\tau$ |
> |- | - | - | - | - | - | - | - |
> | **Ours** 175 tokens constraint | 13.45 | 44.36 | 24.86 | 34.94 | 89.03 | 176 | 9x |
> | Last Sentence | 1.36 | 19.81 | 9.06 | 16.08 | 78.48| 45 | 33x |
>
> Table C. Performance of last sentence baseline with ours in ShareGPT and Arxiv-March23 datasets with GPT-3.5-turbo.
>
> **5. Concerns on GPT4 Generation baseline: “generated prompts omit crucial details from the original prompt... A naive way… would be to tune the prompt of GPT4… What prompt(s) did you use for the compression instruction? Did you tune it/them to make the baseline stronger?”**
>
> We used ten sets of prompts for GPT4 Generation and reported the best results. Examples of prompts used are listed below. In L545-566, we discussed the limitations and shortcomings of the GPT4 generation, including uncontrollable content and text length, difficulty in achieving a high compression ratio due to complete syntax, and high computational costs. We will add these instructions and experimental details to the Appendix.
>
> Example prompts used for GPT4 generation baseline:
>
> 1. Could you please rephrase the paragraph to make it short, and keep 5% tokens?
> 2. Condense the passage to retain only 5% of its original tokens, while preserving its meaning.
> 3. Short the sentences to 200 tokens.
> 4. Trim the text down to 200 tokens in total.
> 5. Please provide a concise summary of the given examples in several sentences, ensuring that all reasoning information is included.
> 6. Summarize the provided examples in a few sentences, maintaining all essential reasoning aspects.
> 7. Remove redundancy and express the text concisely in English, ensuring that all key information and reasoning processes are preserved.
> 8. Eliminate repetitive elements and present the text concisely, ensuring that key details and logical processes are retained.
> 9. Follow these steps to shorten the given text content: 1. First, calculate the amount of information contained in each sentence, and remove sentences with less information. 2. Next, further condense the text by removing stop words, unnecessary punctuation, and redundant expressions. Refine the content while ensuring that all key information is retained. Let's do it step by step.
> 10. To shorten the given text, follow these steps: a) Determine the information value of each sentence and remove those with lower value. b) Further reduce the text by removing stop words, unneeded punctuation, and superfluous expressions, while making sure to keep all vital information intact. Let's do it step by step.
>
> **6. How much compute is required to produce the compressed prompt?**
>
> Please refer to the first response.
>
> **7. How does that affect the latency of the overall system?**
>
> We evaluate the end-to-end inference time of our system. Specifically, it takes about 300ms to compress a 2k-length prompt to a ~400-length prompt ($\tau$=5) using LLaMA-7b in a V100-32G. However, GPT-3.5-Turbo takes about 8.6s to handle the original prompt and about 2.3s to handle the compressed prompt, suggesting that our system can lead to a 3.3x latency speedup.
>
> **8. “If my understanding of the paper is correct, you need roughly #tokens//100 + #demonstrations forward passes in the auxiliary model?”**
>
> The forward passes equal to $1 + k \cdot \frac{L}{\tau \cdot \text{segment\\_size}}$, where $1$ denotes the batch inference carried out by the budget controller on either demonstrations or sentences. $ k \cdot \frac{L}{\tau \cdot \text{segment\\_size}}$ denotes the number of passes in Iterative Token-level Prompt Compression.  But in this scenario, where we compute the perplexity of segment tokens or demonstrations for each forward pass, the computation is the bottleneck. Therefore, it is more reasonable to evaluate the system cost based on the token computation amount (see the response to question 1). Moreover, we also tested the latency of our system, which also showed a realistic speedup (see the response to question 7).
>
> **9: “typos”**
>
> Thank you for your careful reading and suggestions. We will make corresponding corrections.

---

### Official Review · Reviewer_7WkN · 2023-08-05

**Typos Grammar Style And Presentation Improvements:** NA
**Soundness:** 4

**Excitement:**

4: Strong: This paper deepens the understanding of some phenomenon or lowers the barriers to an existing research direction.

**Missing References:**

NA

**Paper Topic And Main Contributions:**

The paper presents LLMLingua, a novel approach to prompt compression for large language models (LLMs). The main contributions include the development of a budget controller, a token-level iterative compression algorithm, and an instruction tuning based method for distribution alignment between language models. The paper showcases the application of these methods on four datasets from diverse scenarios, demonstrating up to 20x compression with minimal performance loss.

**Questions For The Authors:**

Can we compress a longer prompt with more in-context examples into a prompt that fits the input length constraint (e.g. compress 128K tokens into a 4K token input) to potentially enhance downstream task performance?

**Reasons To Accept:**

1. The paper tackles an important problem in LLMs, i.e., the increasing length of prompts, due to chain-of-thought prompting and in-context learning. This work offers a practical solution to accelerate model inference and reduce cost.
2. The proposed approach is novel and incorporates a budget controller to maintain semantic integrity under high compression ratios.
3. The paper provides empirical evidence to support its claims. The experiments conducted on four different datasets demonstrate the versatility and robustness of the proposed method.

**Reasons To Reject:**

1. The authors predominantly concentrate on metrics such as compression rates (referring to the number of tokens in the prompt) and performance loss. However, a more pragmatic and relevant metric would be the end-to-end inference time. In the realm of LLMs, the inference time is typically determined more by the number of tokens generated rather than the number present in the prompt. Consequently, the authors' focus on prompt token reduction may not translate directly to substantial improvements in inference time, which is a key consideration in practical applications.
2. While not necessarily a reason for rejection, the paper does raise an interesting point for discussion: if the prompt can indeed be compressed, does this suggest the possibility of compressing a longer prompt with more in-context examples into a prompt that fits the input length constraint (e.g. compress 128K tokens into a 4K token input) to potentially enhance downstream task performance?

**Reproducibility:**

5: Could easily reproduce the results.

**Reviewer Confidence:**

4: Quite sure. I tried to check the important points carefully. It's unlikely, though conceivable, that I missed something that should affect my ratings.

---

> ### Author Rebuttal · Authors · 2023-08-28
>
> Thank you for your valuable suggestions.
>
> **1: “The authors concentrate on metrics such as compression rates and performance loss. A more pragmatic and relevant metric would be the end-to-end inference time.”**
>
> Our work is primarily focused on black-box LLMs. In this context, we take compression ratio and performance loss as the evaluation metrics for two reasons: (1) the end-to-end latency is influenced by several uncontrollable factors such as network communication and load balancing, which makes it an unsuitable metric for evaluation. (2) LLMs typically charge per token, so compression ratio is an important consideration.
>
> We also evaluate the end-to-end inference time here for your information. Specifically, it takes about 300ms to compress a 2k-length prompt to a ~400-length prompt ($\tau$=5) using LLaMA-7b in a V100-32G. However, GPT-3.5-Turbo takes about 8.6s to handle the original prompt and about 2.3s to handle the compressed prompt, suggesting that our system can lead to a 3.3x latency speedup.
>
> **2. ”raise an interesting point for discussion: can we compress longer prompts like 128K tokens into a 4K token input to potentially enhance downstream task performance?”**
>
> We conducted experiments on the Zero Scrolls dataset (A zero-shot benchmark for long text understanding including question answer, summarization, reasoning tasks). The results show that compressing longer prompts with an average length of 15k to prompts of 4k can achieve better performance than the original 4k-length prompts.
>
> |-|Acc|Tokens|
> |-|-|-|
> |Original Prompt|24.54|4086|
> |**Ours**|30.04|3995|
>
> Table B. The accuracy on Zero-Scrolls validation dataset with GPT-3.5-turbo.

---

### Official Review · Reviewer_89tu · 2023-08-11

**Soundness:** 4

**Excitement:**

4: Strong: This paper deepens the understanding of some phenomenon or lowers the barriers to an existing research direction.

**Paper Topic And Main Contributions:**

This paper proposes a coarse-to-fine prompt compression method, which can compress the prompt within a small performance loss. This paper conducts comprehensive experiments under different compression ratios and different in-context learning tasks and the results show the method is effective. The ablation study also shows the usefulness of different components.

**Questions For The Authors:**

Question A: for distribution alignment, can you give more details about what datasets are used for instruction tuning small LMs? Will the same data be used when evaluating prompt compression?

**Reasons To Accept:**

The paper is well-written. The experiment is comprehensive and shows the proposed prompt compression method is effective under different settings.

**Reasons To Reject:**

It seems perplexity-based token compression method will mainly drop stop words that don't have important semantics. Is it possible that a very simple baseline by just dropping all the stop words while maintaining all the entities can lead to the same performance compared with LLMLingua?

**Reproducibility:**

4: Could mostly reproduce the results, but there may be some variation because of sample variance or minor variations in their interpretation of the protocol or method.

**Reviewer Confidence:**

4: Quite sure. I tried to check the important points carefully. It's unlikely, though conceivable, that I missed something that should affect my ratings.

---

> ### Author Rebuttal · Authors · 2023-08-28
>
> We’re grateful for your insightful suggestions.
>
> **1: “It seems perplexity-based token compression method will mainly drop stop words… a very simple baseline by just dropping all the stop words…”**
>
> The conjecture that our method “mainly drops stop words” is not entirely accurate. In fact, the decision to retain or discard a token in prompt is contingent upon the amount of information it imparts. Stop words may be preserved in cases where they contribute to the logical progression of the text. For instance, in Figure 6, several stop words such as “and”, which separates the reasoning process into clear steps, and 'a', which is used for counting, are retained. Conversely, non-stopwords like “clarinet” and “nectarine” are omitted.
>
> We will incorporate the recommended baseline into Table 3 in the final version to further elucidate the role of stop words in our method. The results are as follows:
>
> |-|EM|Tokens|1/$\tau$|
> |-|-|-|-|
> |**Ours**|79.08|439|5x|
> |w/ remove stop words|76.27|1882|1.3x|
>
> Table A. Ablation study on GSM8K with 1-shot constraint.
>
> **2: “for distribution alignment… what datasets are used for instruction tuning small LLMs? Will the same data be used when evaluating prompt compression?”**
>
> We only use the Alpaca dataset to align the small language models with black-box LLMs, and it is not involved in the evaluation process.
>
> [1] https://github.com/tatsu-lab/stanford_alpaca

---

### Meta-Review · Area_Chair_us5h · 2023-09-17

**Recommendation:** 5

**Metareview:**

The reviewers agree this work demonstrates a well motivated and rigorously tested set of methods for prompt compression, with very promising results! Reviewers indicate their concerns around comparable baselines and evaluation metrics were answered in the rebuttal.

---

### Decision · Program_Chairs · 2023-10-07

**Decision:**

Accept-Main

**Comment:**

The reviewers agree this work demonstrates a well motivated and rigorously tested set of methods for prompt compression, with very promising results! Reviewers indicate their concerns around comparable baselines and evaluation metrics were answered in the rebuttal.